# Morphing Hydrofoil Model Driven by Compliant Composite Structure and Internal Pressure

**Mohammed Arab Fatiha[1,†], Benoît Augier [2,†], François Deniset [1,†], Pascal Casari [3,†] and Jacques André Astolfi [1,*,†,‡]**

1   French Naval Academy Research Institute—IRENav EA3634, 29200 Brest, France;
    fatiha.mohammed_arab@ecole-navale.fr (M.A.F.); francois.deniset@ecole-navale.fr (F.D.)
2   The French Research Institute for Exploitation of the Sea, 29200 Brest, France; benoit.augier@ifremer.fr
3   Research Institute in Civil Engineering and Mechanics—GeM, 44606 Saint-Nazaire, France;
    pascal.casari@univ-nantes.fr
*   Correspondence: jacques-andre.astolfi@ecole-navale.fr; Tel.: +33-298-234-017
†   These authors contributed equally to this work.
‡   This paper is an extended version of our paper published in SMP'19.

**Abstract:** In this work, a collaborative experimental study has been conducted to assess the effect an imposed internal pressure has on the controlling the hydrodynamic performance of a compliant composite hydrofoil. It was expected that the internal pressure together with composite structures be suitable to control the hydrodynamic forces as well as cavitation inception and development. A new concept of morphing hydrofoil was developed and tested in the cavitation tunnel at the French Naval Academy Research Institute. The experiments were based on the measurements of hydrodynamic forces and hydrofoil deformations under various conditions of internal pressure. The effect on cavitation inception was studied too. In parallel to this experiment, a 2D numerical tool was developed in order to assist the design of the compliant hydrofoil shape. Numerically, the fluid-structure coupling is based on an iterative method under a small perturbation hypothesis. The flow model is based on a panel method and a boundary layer formulation and was coupled with a finite-element method for the structure. It is shown that pressure driven compliant composite structure is suitable to some extent to control the hydrodynamic forces, allowing the operational domain of the compliant hydrofoil to be extended according to the angle of attack and the internal pressure. In addition, the effect on the cavitation inception is pointed out.

**Keywords:** smart-structure; hydrofoil; morphing; compliant; composite; cavitation

## 1. Introduction

In naval applications, it is crucial to make strategic decision to reduce the fuel oil consumption of ships and therefore to decrease their $CO_2$ emissions. The demand for the reduction of fuel oil consumption and $CO_2$ emissions is greater than ever before [1]. Underlying the need for improved performance, better comfort, and stability, the use of new concepts of innovative hydrofoils or propeller blades can be an option to enhance the hydrodynamic performance and reduce the consumption of ships.

Using this new concept should allow to control the forces (lift and drag) for various operating conditions to be controlled. However, this can lead to cavitation onset at high speed and moderate angles of incidence but also at low speed and high angles. Improving the hydrodynamic performances and delaying the cavitation inception requires the modification of shape, hence the idea of using morphing hydrofoils.

The aim objective of the current research focuses on aerodynamic and hydrodynamic performance
31 enhancement. In the energy field, we find the works of Aramendia, which analyzed the effect of
Gurney flap (GF) length on the lift/drag ratio $C_L/C_D$ of blades. They have proved numerically the
capability of the GF to improve lift/drag ratio of passive and active flow control [2,3].

Currently, hydrofoils use mechanical systems as a flap to modify their shape and to control
their performance. Morphing structures could be an interesting route to change the hydrofoil
performance [4].

In aerodynamic applications, the use of morphing structures has proved its effect in flying
performance [5]. Brailovski et al. [6] have studied the effect on the aerodynamic performance and
foil mechanical properties of a flexible suction side powered by two actuators numerically. In another
study [7], the gap present at the spanwise ends of the control surfaces is replaced by a smooth,
three-dimensional morphing transition section. The passive control of this compliant morphing flap
transition has the advantage of increasing the lift and reducing the drag. We can not talk about the
benefits of morphing structures on aerodynamic performance, not to mention the effect of various
variable camber continuous trailing edge flap (VCCTEF) on the lift and drag forces [8]. It was noted
that the best stall performance ($L/D$) was demonstrated by the circular and parabolic arc camber flaps.

Even if, the main objectives of the hydrodynamic applications are similar to those of the
aerodynamics, many of the techniques employed in aerodynamic applications cannot be transferred to
naval ones. So, it is necessary to take into account the differences between the fluid properties and the
cavitation phenomena in naval applications.

To meet hydrodynamic requirements, adaptive composites are used in many marine technologies,
including propulsive devices, underwater vehicles, and propellers. In [9], the authors summarized
the progress on the numerical modeling, the experimental studies, design, and optimization of
adaptive composite marine propulsors and turbines. Firstly, they have presented the differences
between adaptive aerodynamic and hydrodynamic lifting surfaces. Therefore in the hydrodynamic
applications, the local pressure fluctuations led to the formation of cavitation [10], which can lead
to load fluctuations, vibrations and performance decay. Furthermore, they discussed the current
challenges in the numerical modeling, experimental studies, design, and optimization of the adaptive
marine propulsors and turbines. The major challenge in the numerical and experimental modeling is
the three-dimensional viscous fluid-structure interaction [11] and the cavitation [12].

The cavitation inception is influenced by several factors. Amromin [13] indicate that the
flow-induced vibration of hydrofoils affects pressure pulsations on their surfaces which depends
on the hydrofoil material and influences the cavitation inception and desinence.

Many of the recent developments have focused on the use of composite materials over
traditional metallic materials. The composite materials have many advantages, including higher
strength-to-weight ratios, better fatigue characteristics, higher durability. They provide resistance to
salt water and improve the resistance to corrosion [14].

The effect of material and Reynolds number on the hydrodynamic performance of hydrofoils
was investigated experimentally by Zarruk et al. [15]. They studied the performances of flexible
hydrofoils of similar geometry made of stainless steel, aluminum and a composite of carbon-fiber
reinforced plastic with layup orientations at 0° and 30°. They concluded that the composite
hydrofoils have the best hydrodynamic performance, showing the potential of a tailored hydroelastic
composite hydrofoil. The hydro-elastic behavior of flexible propellers has also been analyzed by
Maljaars et al. [16]. They compared the results of a boundary element method (BEM) and a
Reynolds-averaged-Navier-Stokes (RANS) simulations with the calculation of measured open water
diagram and the open water curves. They found that the two results of these analyses concurred.

In [17], the cantilevered rigid and compliant three-dimensional hydrofoils were studied in a
cavitation tunnel in order to analyse the cloud cavitation behavior. The rigid hydrofoil was made
of stainless steel and the compliant one of a carbon and glass fiber-reinforced epoxy resin with the
structural fibers aligned along the spanwise direction to avoid material bend-twist coupling. The tests

were carried out at a Reynolds number of $0.7 \times 10^6$, an incidence of $6°$ and cavitation number of 0.8. The compliant hydrofoil was seen to dampen the higher frequency force fluctuations while showing a strong correlation between normal force and tip deflection. Furthermore, the 3D nature of the flow field causes complex cavitation behavior with two shedding modes on both models. Another type of cavitation has been studied by Zhu et al. [18]. They evaluated the hydrodynamic performances of a propeller with winglets numerically and they compared them to those of the benchmark propeller (MAU5-80). They concluded that the presence of the winglets reduces the vapor volume and alleviates the tip vortex cavitation (TVC).

The bend-twist coupling effects on the hydroelastic response of composite hydrofoils have been experimentally studied [19]. The authors concluded that bend-twist coupling affects the deformation of the hydrofoils which modify the hydrodynamic performance. The effect of material bend-twist coupling on the cavitating response of adaptive composite hydrofoils has also been analyzed experimentally [20] for three identical unloaded hydrofoils. Two hydrofoils of composite material and another rigid one of stainless steel (SS) were tested in the same cantilevered configuration. They concluded that material bend-twist coupling has an effect on the hydrodynamic load coefficients, cavitation inception and the maximum cavity length compared to a SS hydrofoil.

In order to assess the effect of the cavitation on the structural response, Ducoin et al. [21] have studied the displacement of a flexible hydrofoil in a cavitating flow. They found that the hydrodynamic loading unsteadiness increases the vibrations experienced by the hydrofoil. Numerically, Garg et al [22,23] have developed a shape optimization tool to predict the hydrodynamic performance including the cavitation inception conditions.

In order to control the lift generated by hydrofoils on boats, Giovannetti et al. [24] have numerically and experimentally analysed hydrofoil geometry designed to reduce the lift coefficient passively by increasing the flow velocity. This study was conducted with the use of wind tunnel experiments including deformation measurements, which concurred with the numerical results. They found that the twist deformations resulted in a reduction in the effective angle of attack by 30% at higher flow velocities, which significantly reduced the foil's lift and drag.

Numerical predictions of the hydrodynamic forces, deformations and cavitation performance for a NACA 0009 hydrofoil and an optimized hydrofoil which have been studied by Garg et al. [22,25] are compared to the experimental ones. The predicted hydrodynamic coefficients ($C_L$, $C_D$, and $C_M$) and the tip bending deflections are concur with measured values for both the baseline and the optimized hydrofoils across a wide range of lift conditions. The mean difference between the numerical predictions and the experimental measurements for mean $C_L$, $C_D$, and $C_M$ for the optimized hydrofoil is noted. This indicates that values are 2.96%, 5.10%, and 3.0%, respectively. The mean difference in the tip bending deflections is 3.45%.

The French Naval Academy Research Institute (IRENav) is interested in the study of the deformed hydrofoils and their performances. Fluid-Structure Interaction has been investigated experimentally by studying the structural response of a flexible lightweight hydrofoil undergoing various flow conditions including cavitating flow by Lelong et al. [26,27]. An optimization of the design of the shape and the elastic characteristics of a hydrofoil equipped with deformable elements giving flexibility to the trailing edge was developed by Sacher et al. [28].

IRENav, the Research Institute in Civil Engineering and Mechanics (GeM) and IFREMER have initiated a research program related to compliant hydrofoils for naval applications. The objective is to characterize a compliant composite hydrofoil driven by an internal low pressure regarding the lift and drag forces as well as cavitation inception. This paper presents the experimental study performed in the hydrodynamic tunnel at IRENav. The hydrofoil manufactured at GeM was firstly tested in the open air to assess the effect of the internal pressure on hydrofoil deformations by the use of the digital image correlation (DIC-3D) system. Then, the hydrofoil was tested in the cavitation tunnel at IRENav where the lift and drag coefficients and the hydrofoil deformations were measured. In accordance with the experiments, a numerical approach based on a fluid-structure coupling algorithm has been

developed. The paper describes the experimental setup, the numerical fluid-structure interaction (FSI) algorithm and presents the main results.

## 2. Experimental Setup

The experiments are carried out in the cavitation tunnel at IRENav (Figure 1). The tunnel test section is 1 m long with a square section of 0.192 m side (Figure 2). The inflow velocity ranges between 0.5 and 12 m/s. The pressure in the tunnel test section ranges from 100 mbar and 3 bar to control the cavitation which is given by a cavitation number defined by Equation (1) and the measured turbulence intensity in the test section is 2% at 5 m/s. This cavitation number can therefore be compared with the opposite of pressure coefficient $-C_{pmin}$ defined as the minimum of the pressure coefficient (Equation (2)).

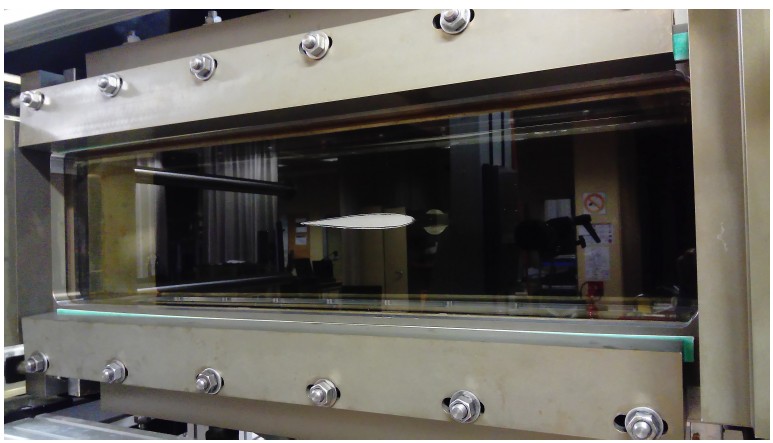

**Figure 1.** Hydrodynamic tunnel test section at IRENav with the compliant composite hydrofoil.

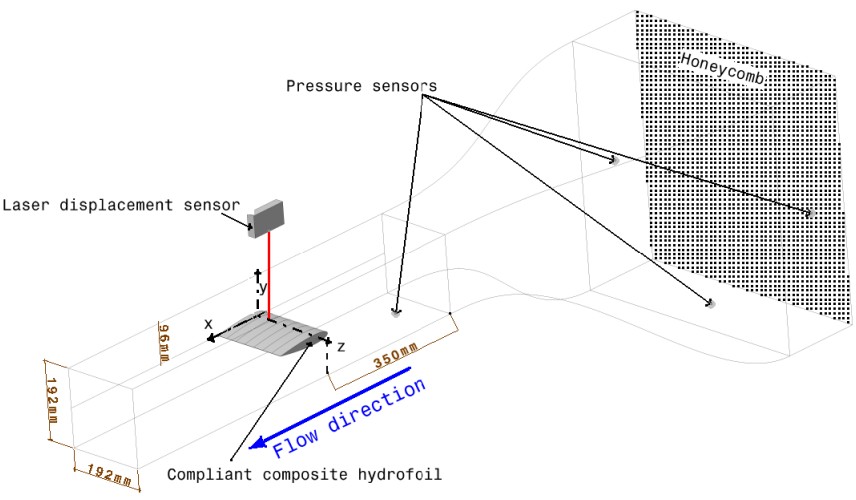

**Figure 2.** Tunnel test section characteristics with the compliant composite hydrofoil and Laser displacement measurement system.

$$\sigma = \frac{P_{ref} - P_v}{\frac{1}{2}\rho V^2} \tag{1}$$

$$C_p = \frac{P - P_{ref}}{\frac{1}{2}\rho V^2} \tag{2}$$

where $P_{ref}$ is the pressure in the test section, $P_v$ is the vapor pressure at the water temperature, $P$ is the local pressure, $V$ is the inflow velocity, and $\rho$ is the water density. Thus, when $\sigma < -C_{pmin}$, that is to say when $P < P_v$, cavitation is expected to appear in the flow at the point where the pressure coefficient is the lowest.

The compliant composite hydrofoil was manufactured at the Research Institute in Civil Engineering and Mechanics (GeM). At rest, the compliant composite hydrofoil has a NACA 0012 section and a rectangular plan-form of 0.191 *m* span and 0.15 *m* chord length. It is cantilevered and clamped on a cylindrical aluminum beam fitted to the hydrodynamic balance. The axis of rotation is at $X/c = 0.25$.

The hydrofoil is composed of two walls, one rigid and one compliant, providing a cavity in which vacuum can be applied to vary the shape. In the following part, pressure variation will always correspond to a suction inside the hydrofoil compared to the atmospheric reference in the test section. It is defined as $\Delta P$ (called internal pressure in the paper) and has a positive value as it decreases. The pressure system, the compliant composite hydrofoil and the pneumatic actuator are presented in Figure 3. The pressure inside the cavity is measured using a manometer.

The compliant wall is laminated with three carbon/epoxy plies oriented at $0°/90°$ in the middle, and thinner at the leading and trailing edges. These two edges are composed of two plies: one of carbon/epoxy $[0°/90°]$ and the second of glass fiber with an orientation at $45°$. The rigid side is composed of five plies of carbon/epoxy with the layups orientations at $0°$ and $90°$ (Figure 4).

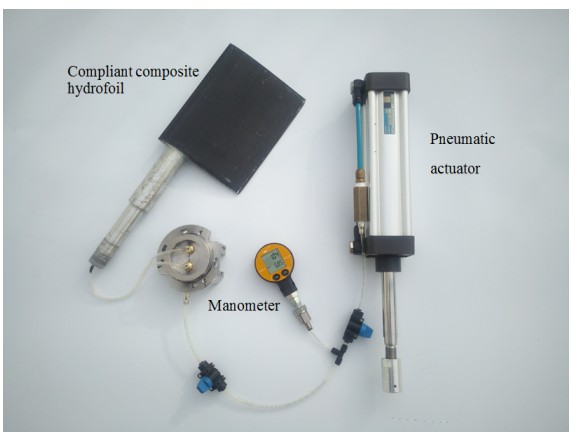

**Figure 3.** Compliant composite hydrofoil equipped with the control internal pressure system.

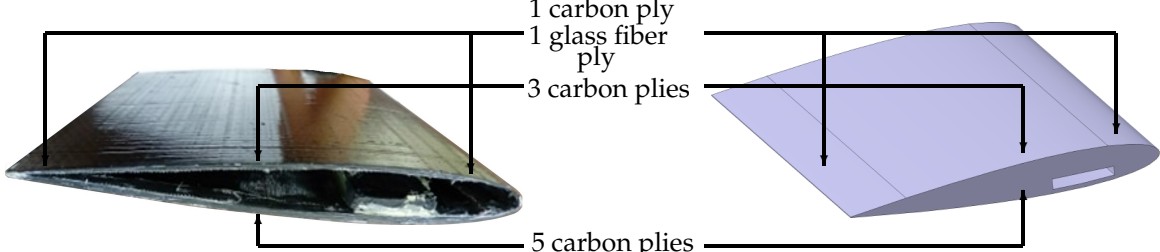

**Figure 4.** Laminate structure of the compliant composite hydrofoil.

To assess the effect of the internal pressure on hydrofoil deformations in open air, the chordwise displacement of skin is investigated using the digital image correlation (DIC-3D) system at GeM laboratory (Figure 5). The digital image correlation is well known as an effective method of obtaining field surface displacements. This method is based on optimal strain measurements. It is a non-contact technique and it is particularly suitable for flexible materials. It is based on the comparison of two digital images features of the composite surface before and after loading, and total displacement and

strain fields can be obtained. For this purpose, a two digital camera system has been used to monitor the strain pre-gressing on the surface and a computer with DIC software. In order to produce fine and exploitable details, a random pattern of paint is usually applied to the surface of the hydrofoil (Figure 6). The software selects points in the reference image and follows them in the following images thanks to a window defined around the points. The window consists of some pixels which grant a unique greyscale intensity distribution to the window. To detect the area with the most similar intensity distribution, which contains the required points, a cross-correlation method is used to scan the next image. Thus, the movements of two separate points are followed by DIC and the last computes the change in the distance between these points [29].

Compliant hydrofoil

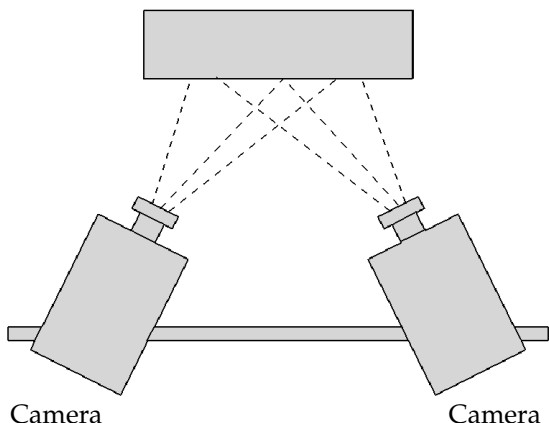

Camera                    Camera

**Figure 5.** Digital image correlation (DIC) schematic at GeM (Saint-Nazaire, France).

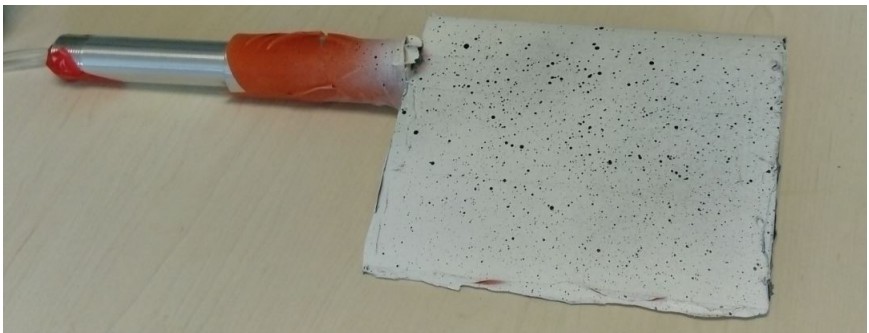

**Figure 6.** Specimen prepared for DIC with random pattern paint on the surface.

The hydrofoil displacement is also investigated in open air by using a micrometric touch probe at a discrete position ($Z/c = 0.33$, $X/c = 0.63$) for various imposed internal pressures of $\Delta P = 0.15$ bar, 0.4 bar and 0.51 bar.

In the hydrodynamic tunnel, the static deformation is measured using a Laser distance measurement system mounted on a 2D translation system on the upper side of the test section. The system measures the vertical position of the hydrofoil suction side along sections scanned through the span from the root to the tip. It continuously monitors displacement according to a sampling frequency chosen by the user (set to 50 Hz). The system allows us to scan the hydrofoil surface for a given flow condition along various sections selected along the span. In this case, nine sections from the root to the tip are selected. At first and for the $\Delta P^* = 0$, the distance measurements are carried out without a flow, in order to determine the rest position of the hydrofoil. Measurements are repeated under a flow at 5 m/s. The results of the laser distance measurements are presented in Figure 7. It is shown that the velocity has no effect on the hydrofoil displacement. After, for the same inflow velocity,

the distance measurements are carried out for various internal pressures. The deformation is obtained by comparing the scans between referenced and tested internal pressures.

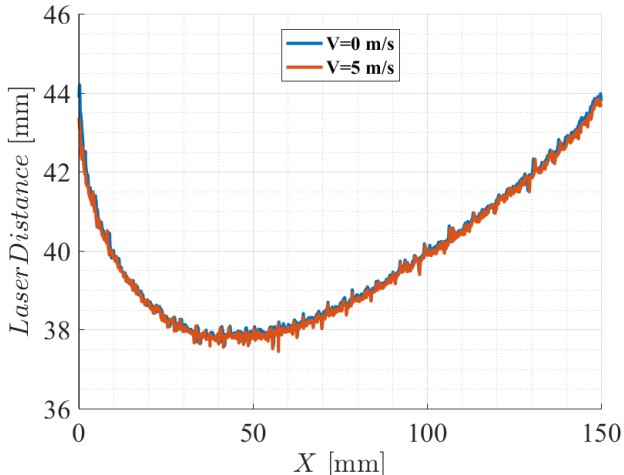

**Figure 7.** Results of the laser displacement measurement system without flow and $Re = 0.75 \times 10^6$.

In the hydrodynamic tunnel, measurement of hydrodynamic forces is conducted using a hydrodynamic balance at various conditions of internal pressure and angle of attack and different flow velocities. Firstly, the hydrodynamic forces are measured for a velocity of 5 m/s corresponding to a Reynolds number of $0.75 \times 10^6$. Secondly, they are measured for a velocity of 6.67 m/s and 9 m/s which correspond to $10^6$ and $1.35 \times 10^6$ Reynolds numbers respectively. This increase in velocity is in order not to increase the incidence too much and in order to analyze the cavitation inception for low cavitation numbers.

The 5-component hydrodynamic balance has a range of up to 1700 N for the lift force, 180 N for the drag and a 43 N m for the pitching moment. It is fixed into a supporting frame, mounted on bearings, and driven in rotation by a Baldor motor. The stepper motor allows for 600,000 impulsions per rotation, meaning a resolution of $6 \times 10^{-4\circ}$. The foil is fastened into the balance, secured by a tightly fitted key/nut system [30]. As the test section is horizontal, the geometric $0°$ angle of attack of the hydrofoil is visually controlled using the water surface at mid height of the test section when filling the tunnel. Furthermore, as the hydrofoil is symmetric, the zero-lift angle is used for the positioning of the final angle of attack.

## 3. Uncertainties

The experimental uncertainties consist the precision of the hydrodynamic balance, pressure sensors, Laser distance measurement system, digital image correlation (DIC-3D) and micrometric touch probe.

In the cavitation tunnel, the uncertainties of velocity and pressure measurements are based on the accuracy of the pressure sensors. The latter is about 0.04 bar. For the measurements of hydrodynamic forces and from the document provided by the manufacturer of the hydrodynamic balance, the uncertainties are about $\pm 1.02$ N for the lift, $\pm 0.324$ N for the drag and $\pm 0.26$ N.m for the pitching moment. The uncertainty on the displacement measurements in the test section is about $\pm 0.046$ mm.

For open air measurements, the uncertainty of the digital image correlation (DIC-3D) system is about $\pm 0.002$ mm for the in-plane displacements and $\pm 0.004$ mm for the displacements out of the plane. The micrometric touch probe precision is about $\pm 0.01$ mm.

## 4. Numerical Approach

The numerical study consists of 2D simulation to investigate the effect of a static internal pressure on the structural response of the compliant hydrofoil as well as the impact on hydrodynamic performances.

The flow model of the XFOIL solver and the finite-element method of the ANSYS-Mechanical solver are used for the FSI analysis. The FSI algorithm is based on an iterative method between the two solvers with a small perturbation hypothesis. The flow model is based on the coupling between a panel method with a boundary layer model. More details concerning Xfoil are given in [31]. The panel method accelerates the flow calculations as compared to finite volume methods.

The numerical model of the hydrofoil structure is calculated by ANSYS Mechanical using ANSYS APDL (Ansys Parametric Design Language) with solid elements according to the preliminary hydrofoil design. The original APDL script was modified to handle distinct geometry changes and meshing. Solid elements PLANE183 (quadratic) are used to mesh the hydrofoil geometry.

The coupling algorithm is developed using Python-scripts. The fluid-structure coupling algorithm is described in Figure 8. The FSI algorithm is initialized by a structural computation as the cantilevered hydrofoil is submitted to an internal pressure only. It leads to displacements which produce to a new hydrofoil shape. Then, the viscous flow around the new foil is solved. The computation returns a $C_p$ distribution and the forces of the hydrofoil. The external hydrodynamic pressure resulting from the $C_p$ distribution is applied during the structural resolution. The problem is solved using an iterative method until the convergence on the maximum structural displacement and the lift coefficient $C_L$ is reached.

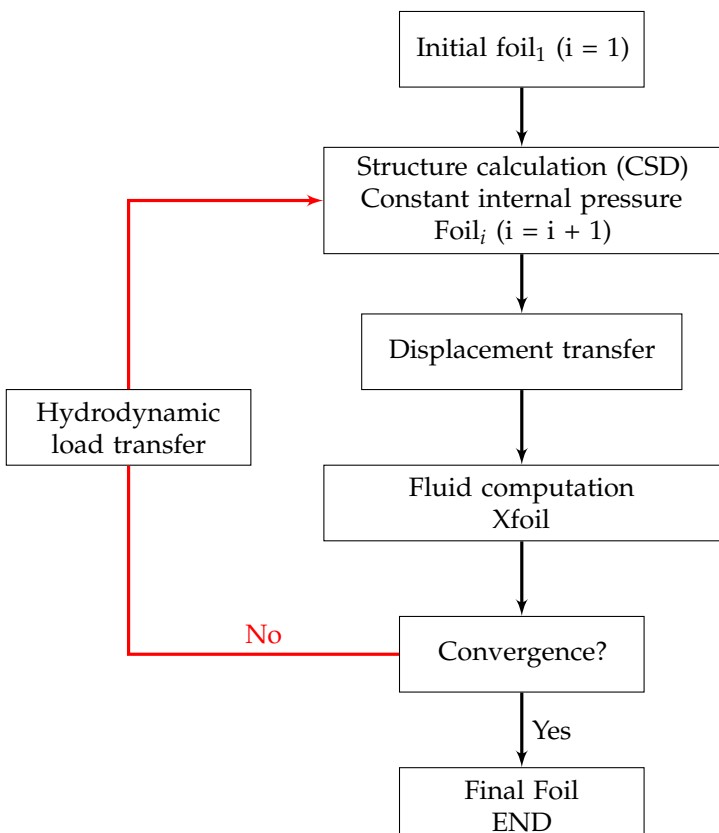

**Figure 8.** Schematic algorithm of fluid-structure coupling with imposed internal pressure.

The convergence to the equilibrium of the hydrofoil is obtained after a small number of iterations showing that the method developed in this work has an advantage when compared to advanced CFD-CSD solvers that require very significant CPU (Central Processing Unit) times.

In a first approach and to simplify the calculation, the hydrofoil material is considered as an homogeneous elastic equivalent. The Young's modulus input for the 2D section of the compliant hydrofoil was adjusted until the maximum displacement of the hydrofoil in the simulations coincided with the maximum displacement of the hydrofoil during the experiments for various internal pressures. The equivalent Young modulus used in the computation was set to $E = 70,000$ MPa and the equivalent Poisson coefficient was set to 0.34.

The initial and deformed shapes of the hydrofoil at a 3° angle of attack, $Re = 0.75 \times 10^6$ and an imposed internal pressure of $\Delta P^* = 1.92$ are presented if Figure 9. The calculated maximum chordwise displacement of the hydrofoil was found to be 3.3 percent of the chord length.

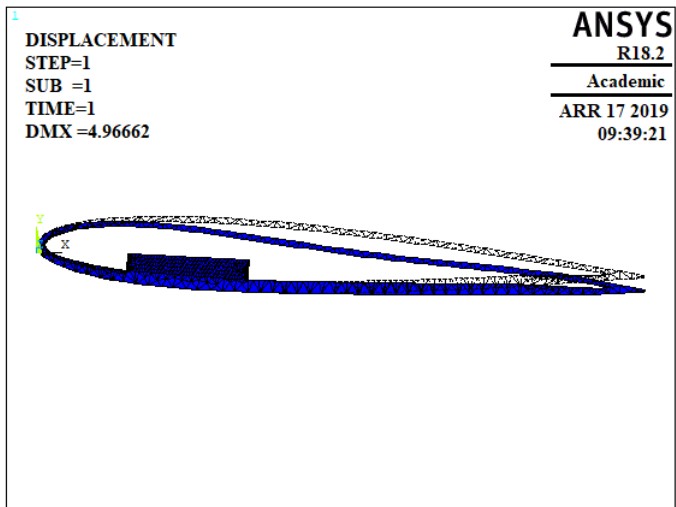

**Figure 9.** Initial shape and deformed shape of the hydrofoil at the first iteration, $\Delta P^* = 1.92$, $\alpha = 3°$ and $Re = 0.75 \times 10^6$.

## 5. Results and Discussion

### 5.1. Hydrofoil Deformation and Hydrodynamic Forces

The effect of internal pressure on the hydrofoil deformation is investigated by using digital image correlation (DIC-3D). For $\Delta P = 0.415$ bar, the displacement field plotted against the foil coordinates is presented in Figure 10. It is observed that the maximum displacement is 8.06 mm (5.3%$c$) located at the center of the hydrofoil. The deformation is not uniform along the spanwise direction due to the structural boundary conditions at the root and the tip.

The displacement of the hydrofoil as a function of the chord is taken at $Z/c = 0.55$ for four internal pressures. It is plotted in Figure 11. It confirms the results of the 3D displacement field presented in Figure 10. The hydrofoil displacement is not uniform along the chord which explains the modification of the camber.

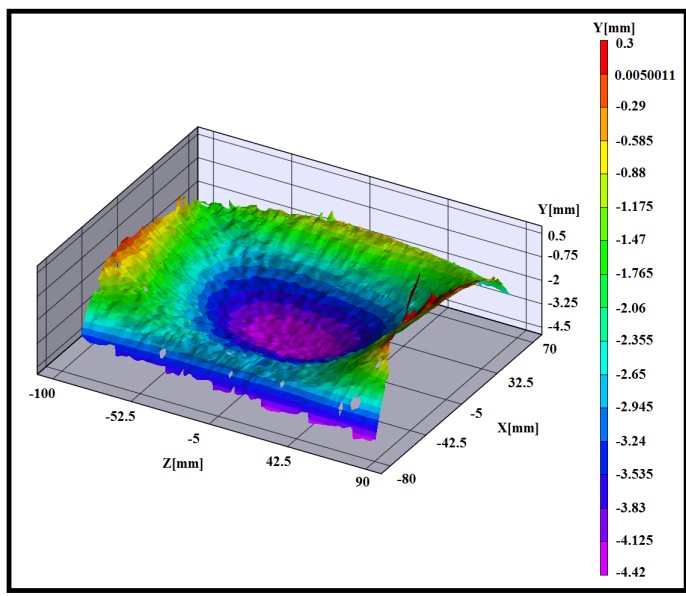

**Figure 10.** DIC-3D displacement, $\Delta P = 0.415$ bar.

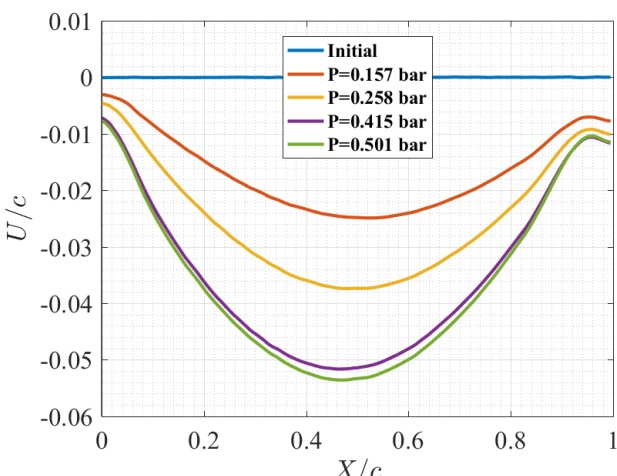

**Figure 11.** Experimental hydrofoil displacement in open air plotted against the chord and the internal pressure at $Z/c = 0.55$.

The hydrofoil displacements are also measured with a micrometric touch probe at $X/c = 0.33$ and $Z/c = 0.63$. The results are compared to the numerical simulation and to the measurements of the DIC-3D system at the same point (Figure 12). As shown in Figure 10, the displacement is linear for an internal pressure of up to $\Delta P = 0.4$ barand reaches a limit value of about $0.045c$. The numerical results correspond to experimental ones for an internal pressure of up to $\Delta P = 0.4$ bar showing that the equivalent homogeneous model is fairly consistent with the experiment. Beyond $\Delta P = 0.4$ bar, the predicted deformation increases linearly exhibiting no saturation.

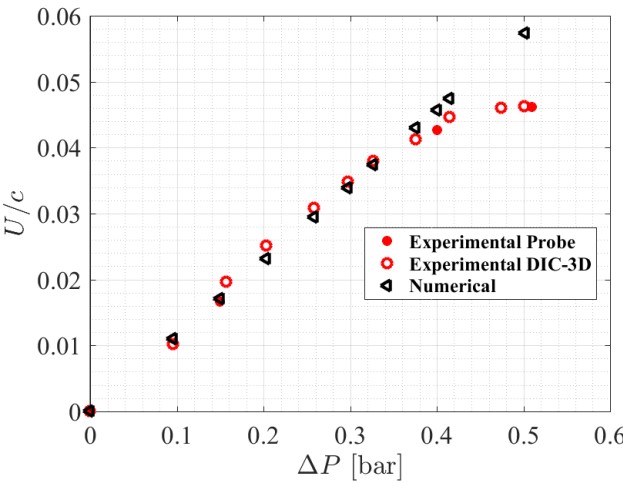

**Figure 12.** Experimental and numerical displacement in open air at $Z/c = 0.33$ and $X/c = 0.63$.

The foil deformation is then analyzed using the scanning measurement system in the hydrodynamic tunnel for a velocity of 5 m/s. The experimental deformed section is presented in Figure 13 for an internal pressure $\Delta P^* = 1.92$ and compared to the numerical one. As depicted in Figure 13, the experimental and numerical displacements have the same trend excepted at the trailing edge where a significant difference is observed. This difference is due to the twist of the hydrofoil in the experiment that is not observed with a 2D computation in the numerical study. The connection between the core beam (Figure 9) and the low pressure side skin is indeed flexible and allows the whole section to rotate around the cantilevered one.

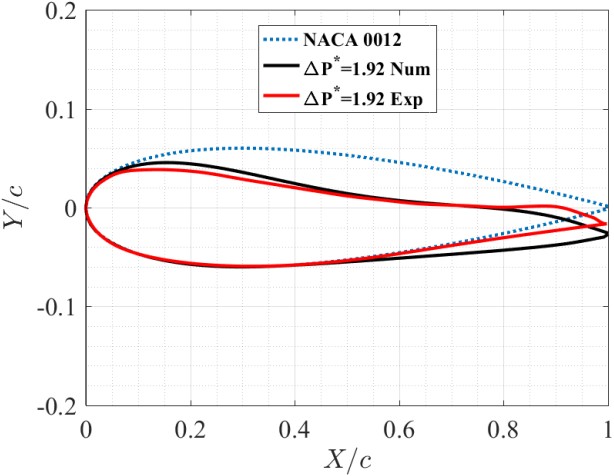

**Figure 13.** Experimental and numerical foil sections, $\Delta P^* = 1.92$ and $V = 5$ m/s.

The hydrodynamic forces for different internal pressures are presented in Figure 14a–c. It is reminded that when the pressure inside the cavity decreases, $\Delta P^*$ increases. It is shown that the lift coefficient and lift/drag ratio shift upwards as $\Delta P^*$ increases with a slight change in the slope. For the ranges of angles of attack and internal pressure of the present experiments, it is found that the lift coefficient and lift/drag ratio increase linearly for both parameters. The results of Figure 14a can be explained by the response surface of the compliant hydrofoil in terms of lift coefficient versus angle of attack and the non-dimensional internal pressure. The operating domain of the compliant hydrofoil is plotted as a function of the two independent variables in Figure 15.

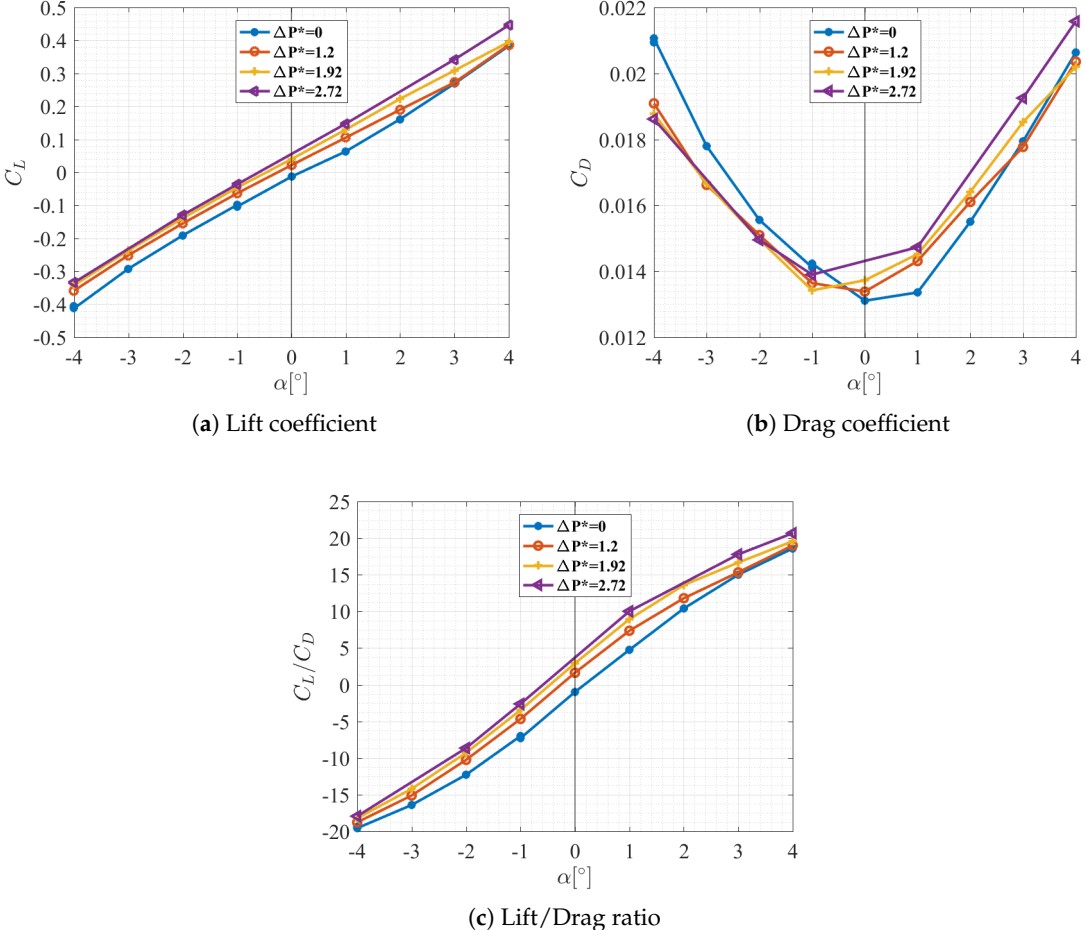

**(a)** Lift coefficient

**(b)** Drag coefficient

**(c)** Lift/Drag ratio

**Figure 14.** Experimental lift and drag coefficients as a function of angle of attack and an internal pressure at $Re = 0.75 \times 10^6$.

According to theory, the lift coefficient depends on the maximum thickness and angle of attack (Equation (3)). In this study, it depends on maximum thickness, angle of attack and internal pressure (Equation (4)).

$$C_L = 0.109(1 - k\tau)\alpha \tag{3}$$

$$C_L(\alpha, \Delta P^*) = C_L(\alpha, \Delta P^* = 0) + \Delta C_L(\alpha, \Delta P^*) \tag{4}$$

If it is considered that the internal pressure changes the thickness (see Figure 12) and the camber linearly, the maximum thickness and $\Delta C_L$ depend only on internal pressure variation. A linear approximation of the response surface (Figure 15) is shown in Equation (5).

$$C_L(\Delta P^*, \alpha) = 0.109(1 - k(\tau + a\Delta P^*))\alpha + b\Delta P^* \tag{5}$$

where $\Delta P^* = \frac{\Delta P}{q}$ is the non dimensional internal pressure and $a = 0.013$, $b = 0.025$ and $k = 0.95$ are constants, which depend on material and hydrofoil design, determined from linear regressions on experimental data.

This simple expression highlights the effect of the internal pressure on the lift slope and lift coefficient described experimentally. The latter is equivalent to the shift of the zero lift angle as a result of camber modification. The expression of the surface response depending on flow conditions and design parameters can be very useful in the context of foil optimization methods. The same approach could be used regarding the drag coefficient and the lift to drag ratio.

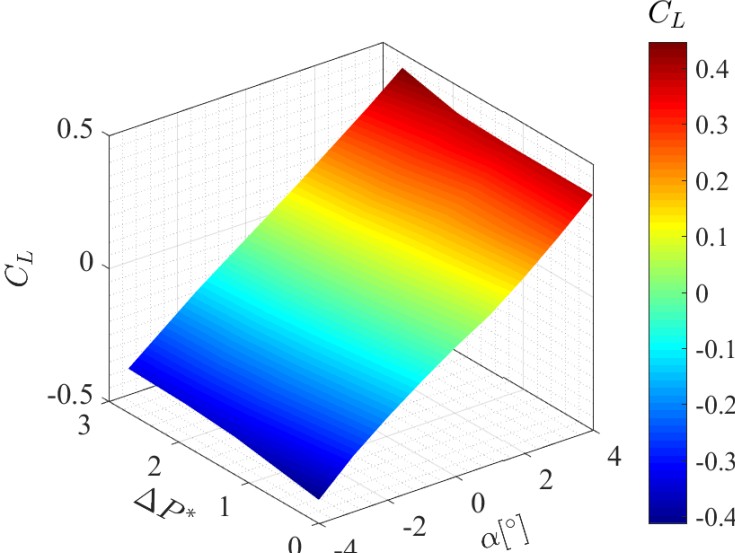

**Figure 15.** Experimental surface response of the lift coefficient plotted against the angle of attack and internal pressure at $Re = 0.75 \times 10^6$.

The experimental and numerical lift coefficients are presented in Figure 16. The numerical lift coefficient obtained from the FSI algorithm concurs with the experimental one but departs progressively from the experiment as the angle of attack progressively increases. The reason fro the discrepancies can be found in the analysis of the structure. Indeed, the experimental section shapes and displacements of the compliant wall are extracted at mid-span for $\alpha$ ranging from $-4°$ to $4°$ and they are shown in Figure 17a,b for the positive and negative angles of attack respectively.

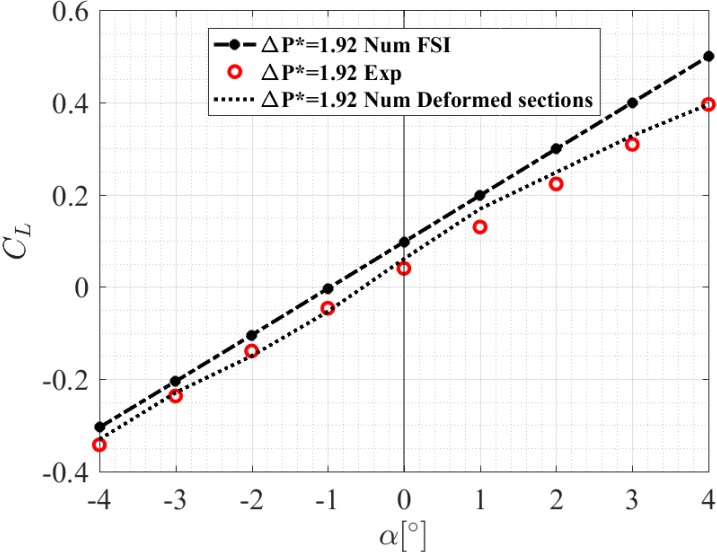

**Figure 16.** Experimental and numerical lift coefficients as a function of the angle of attack for an internal pressure $\Delta P^* = 1.92$ and $Re = 0.75 \times 10^6$. FSI computation and flow computation over the experimentally deformed sections at mid-span.

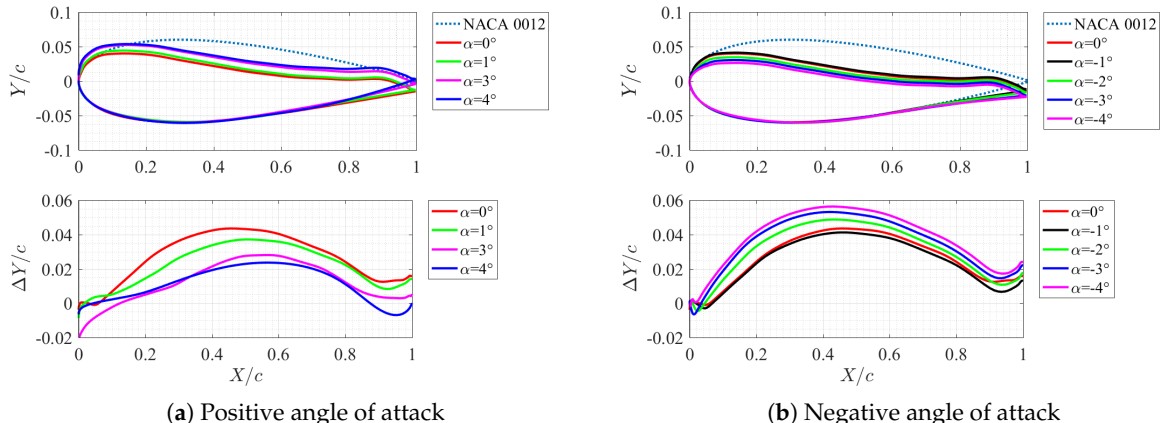

**Figure 17.** Experimental mid-span shapes and associated displacements for $\Delta P^* = 1.92$, $Re = 0.75 \times 10^6$ and positive and negative angles of attack.

A close examination shows that a small flap effect is observed experimentally at the trailing edge affecting the lift coefficient. This effect is not shown by the 2D structural solver where the connection between the skin and the structural beam is considered as rigid. Moreover, experimentally the whole section rotates around 0.25 $X/c$ due to a slight twisting of the structural internal beam as previously described. This is particularly observed for a positive angle of attack. The foil twisting tends to reduce the angle of attack, therefore to decrease the lift coefficient. It is observed that the FSI computation concurs very well the experiments for negative angles of attack when the twist can be neglected. Furthermore, for positive angles of attack, the twist observed experimentally is not seen into the FSI simulation. This is clearly shown in Figure 16 where the lift coefficient is computed directly on the experimental deformed sections at mid-span. In this case, the numerical lift concurs well the experiments all over the angle of attack range. The following parameters should be explored as they can impact the numerical-experimental comparison as a structural model: geometry, material properties (isotropic or orthotropic), structural boundary conditions, flow 3D effects and confinement in the test section.

### 5.2. Cavitation Control

In addition, the effect of the internal pressure on cavitation is analyzed. The effect of the internal pressure on the theoretical cavitation inception is numerically predicted using the FSI algorithm. The lift coefficient versus the opposite of the minimum pressure coefficient ($-C_{pmin}$) for the compliant hydrofoil under various internal pressures is shown in Figure 18. The internal pressure has a direct influence on the theoretical cavitation inception, particularly for the lift coefficients above $-0.1$.

Experimentally, the cavitation inception for an internal pressure equal to $\Delta P^* = 3.1$, an angle of attack $\alpha = 7.4°$ and a cavitation parameter $\sigma = 3.8$ at a Reynolds number of $10^6$ is shown in Figure 19a. For identical conditions, it can be shown that cavitation disappears (Figure 19b) by only decreasing the internal pressure to $\Delta P^* = 1.71$. The corresponding hydrofoil shapes at cavitation inception and desinence is shown in Figure 20. This shape modification of the hydrofoil leads to a slight decrease in the lift coefficient from 0.805 to 0.754.

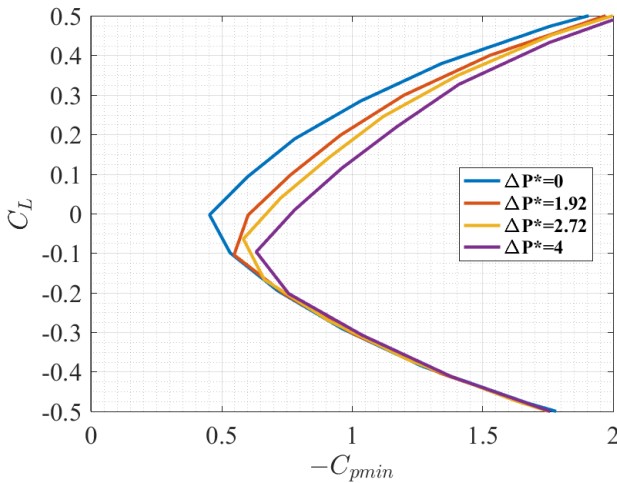

**Figure 18.** Numerical cavitation bucket, $Re = 0.75 \times 10^6$, $\Delta P^* = 0$, $\Delta P^* = 1.92$, $\Delta P^* = 2.72$ and $\Delta P^* = 4$.

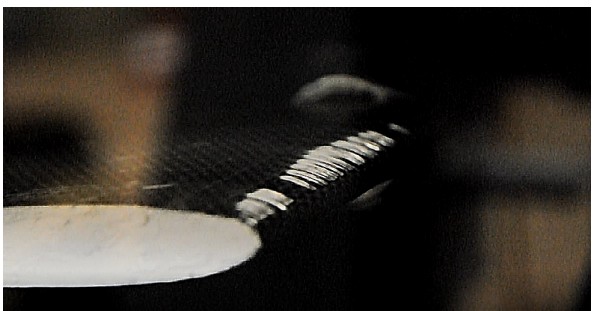

(**a**). Cavitation inception
$C_L = 0.805$
$\Delta P^* = 3.1$

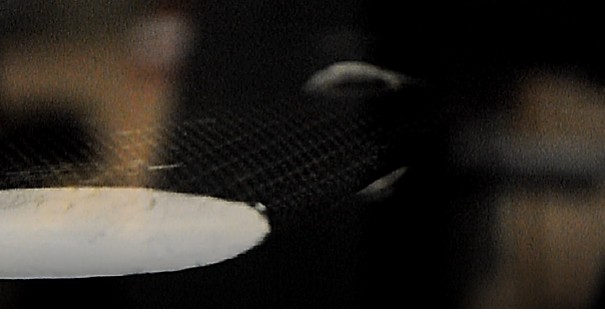

(**b**). Cavitation desinence
$C_L = 0.754$
$\Delta P^* = 1.71$

**Figure 19.** Cavitation inception and desinence on a compliant composite hydrofoil, $Re = 10^6$, $\alpha = 7.4°$ and $\sigma = 3.8$.

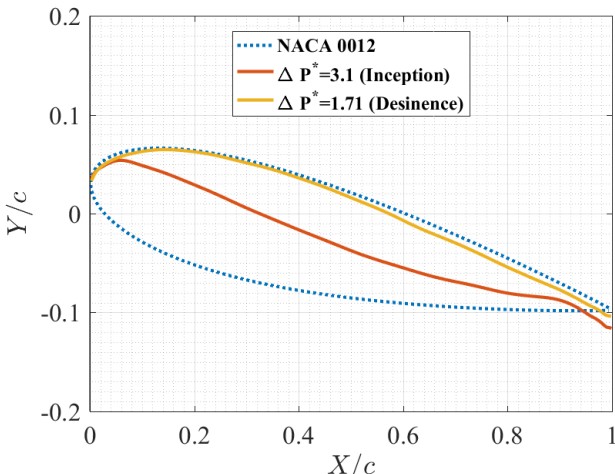

**Figure 20.** Experimental hydrofoil shapes at cavitation inception and desinence at $Re = 10^6$, $\alpha = 7.4°$ and $\sigma = 3.8$.

From the theoretical cavitation bucket (Figure 18), it is noted that the internal pressure has an effect on the cavitation inception for a positive lift coefficient. For this reason, the numerical and the experimental cavitation buckets are compared for a Reynolds number $Re = 1.35 \times 10^6$, positive lift coefficient and two internal pressures. The numerical and experimental cavitation buckets of a the symmetrical hydrofoil and a deformed one under internal pressure $\Delta P^* = 0.59$ are presented in Figure 21. It is shown that the numerical results concur with the experimental ones for the range of lift coefficient of the present analysis. Furthermore, it can be noted that the internal pressure has an effect on the non-cavitation domain. The differences between the numerical and the experimental results could be explained by some pressure fluctuations in the test section.

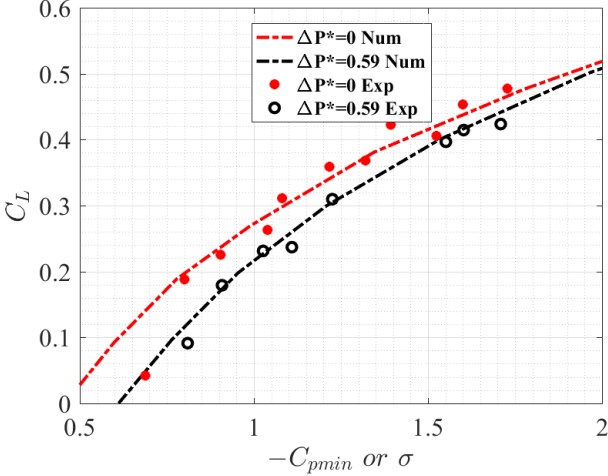

**Figure 21.** Experimental and numerical cavitation buckets of a symmetrical hydrofoil and a deformed one under internal pressure $\Delta P^* = 0.59$, $Re = 1.35 \times 10^6$.

## 6. Conclusions

In this paper, an experimental and numerical study has been presented in order to assess the effect of the internal pressure on the hydrodynamic performance of a compliant composite hydrofoil. The compliant hydrofoil was controlled by a cavity driven by pressure (suction). It was tested in a cavitation tunnel at $Re = 0.75 \times 10^6$ at different angles of attack. Firstly, the hydrofoil deformations were measured in open air using digital image correlation (DIC-3D) and a micrometric touch probe.

Secondly, experiments were performed in a hydrodynamic tunnel where lift and drag coefficients were measured using a hydrodynamic balance together with the compliant skin deformation using a scanning laser measurement system.

A 2D numerical fluid-structure coupling is developed in accordance with the experiments. It was based on an iterative method coupling the flow solver Xfoil and the structural solver ANSYS-Mechanical. Experiments and simulations were carried out at different angles of attack and various imposed internal pressures.

The experiments show that internal low pressure variation has a significant effect on the hydrofoil shape and thus on the hydrodynamic forces. Experimentally, the internal low pressure variation leads to section variations (thickness, camber) together with a slight overall twisting and a small flap effect at the trailing edge. The increase of the suction in the cavity tends to increase the lift coefficient and the lift/drag ratio but at the expense of increased minimum pressure and hence the incipient cavitation number. It is also shown that cavitation can be controlled to some extent by changing only the internal pressure for a given angle of attack and a given flow velocity. The increase of the suction in the cavity tends also to decrease the slopes of the lift coefficient and the lift/drag ratio. A response surface depending on the the internal pressure and the angle of attack can be simply derived from a linear approximation.

The numerical cavitation bucket predicted from the FSI algorithm was compared to the experimental one and the results correlated. The morphing hydrofoil model developed in this work based on a compliant composite structure driven by an internal pressure has provided encouraging results. It allows the hydrodynamics forces and the cavitation to be controlled, paving the way for optimization methods to enhance hydrodynamic performances based on Fluid-Structure Interactions.

**Author Contributions:** M.A.F. carried out the experiments, developed the numerical tool with a help from F.D. for the whole software part, and wrote the manuscript under the supervision of all authors. P.C. manufactured the hydrofoil and characterized it with digital image correlation. J.A.A. set up the different measurement systems at IRENav (cavitation tunnel with the laser distance measurement system and the hydrodynamic scale with a help from B.A.). All authors discussed the results and commented on the manuscript.

**Funding:** This research received no external funding.

**Acknowledgments:** The authors would like to thank the technical staff of the French Naval Academy Research Institute, as well as Research Institute in Civil Engineering and Mechanics (Saint-Nazaire) for their support to this study.

**Conflicts of Interest:** The authors declare no conflict of interest.

## Nomenclature

| | |
|---|---|
| CFD | Computational Fluid Dynamics. |
| CSD | Computational Structural Dynamics. |
| FSI | Fluid-Structure Interaction. |
| $\alpha$ | angle of attack [°]. |
| $\epsilon$ | convergence criteria. |
| $\rho$ | fluid density [$kg/m^3$]. |
| $\sigma$ | cavitation number: $\sigma = \frac{P - P_v}{\frac{1}{2}\rho V^2}$. |
| c | hydrofoil chord [m]. |
| $C_D$ | drag coefficient: $C_D = \frac{D}{\frac{1}{2}\rho V^2 s}$. |
| $C_L$ | lift coefficient: $C_L = \frac{L}{\frac{1}{2}\rho V^2 s}$. |
| D | drag force [N]. |
| $\Delta P = P_{atm} - P_{internal}$ | difference between the atmospheric reference and internal pressure [bar]. |
| $\Delta P^* = \Delta P/q$ | non dimensional internal pressure. |

| e | hydrofoil span [m]. |
|---|---|
| L | lift force [N]. |
| P | pressure [bar]. |
| q | dynamic pressure: $q = \frac{1}{2}\rho V^2$. |
| Re | Reynolds number: $Re = Vc/\nu$. |
| s | hydrofoil planform. |
| U | total displacement [m]. |
| V | inflow velocity [m/s]. |
| $X, Y, Z$ | foil coordinates [m]. |
| $\nu$ | kinematic viscosity [m$^2$/s]. |
| $\tau$ | maximum thickness [m]. |

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
