# Peer review of "Morphing Hydrofoil Model Driven by Compliant Composite Structure and Internal Pressure"

_jmse, doi:10.3390/jmse7120423_

Round 1
Reviewer 1 Report
I have read carefully the article titled: "Morphing Hydrofoil Model Driven by Compliant Composite Structure and Cavity Pressure", article ID: jmse-639363. The paper discusses experimental and numerical investigations on a compliant/deformable hydrofoil under different conditions (angle of attack, cavitation etc.).
The paper is interesting, well presented and the language is adequate (though minor mistakes exist, so I would suggest the authors to proof-read their manuscript).
I have only minor comments to make:
-The authors present an experimental aspect of their work. They should detail experimental errors and uncertainties, as these are crucial to understand their influence on CFD/experimental discrepancies.
- The authors used XFOIL to analyse the flow, which is a potential flow code. It has corrections that can capture viscous effects, up to a point, though complex 3D effects (see also their comment at ln. 256-257), such as turbulence and separation cannot be captured. Indeed it is faster than e.g. Finite Volumes, but is it accurate in all cases? The authors should comment on the role of the assumptions of their models on the errors that have been found.
-The adjustment of hydrofoil mechanical properties in lines 226-229 seems a bit like adjusting to achieve the desired experimental results. Why the actual mechanical properties have not been used?
Author Response
Point 1: The paper is interesting, well presented and the language is adequate (though minor mistakes exist, so I would suggest the authors to proof-read their manuscript).
Response 1: The english has been checked by a native English colleague.
Point 2: The authors present an experimental aspect of their work. They should detail experimental errors and uncertainties, as these are crucial to understand their influence on CFD/experimental discrepancies.
Response 2: The uncertainties of velocity and pressure measurements are based on the accuracy of the pressure sensors. The latter is about 0.04bar. For the measurements of hydrodynamic forces and from the document provided by the manufacturer of the hydrodynamic balance, the uncertainties are about 1.02N for the lift, 0.324N for the drag and 0.26N.m for the pitching moment. The uncertainty on the displacement measurements in the test section is about 0.046mm.
In open air, the uncertainties of the displacements measurements are about 0.002mm in-plane and 0.004mm out of the plane for the DIC-3D and 0.01mm for the micrometric touch probe.
The text has been added in the article.
Point 3: The authors used XFOIL to analyse the flow, which is a potential flow code. It has corrections that can capture viscous effects, up to a point, though complex 3D effects (see also their comment at ln. 256-257), such as turbulence and separation cannot be captured. Indeed it is faster than e.g. Finite Volumes, but is it accurate in all cases? The authors should comment on the role of the assumptions of their models on the errors that have been found.
Response 3: The Xfoil model is only used in 2D cases in this study and only small angles of attack are considered. This limitation is due to the coupling between the inviscid and viscous models. Xfoil gives accurate results for Reynolds numbers and angles of attack considered in this study.
Point 4: The adjustment of hydrofoil mechanical properties in lines 226-229 seems a bit like adjusting to achieve the desired experimental results. Why the actual mechanical properties have not been used?
Response 4: The tested foil with internal pressure is made of a composite structure. The laminate in the structure has different mechanical properties in the section due to different numbers of plies and layers properties. The authors have chosen, in a first approach to consider the foil as made of a homogeneous materiel to simplify the calculation. The properties of the “equivalent material” have been tuned accordingly.
The text has been changed in the article for more clarity.

Reviewer 2 Report
The authors present interesting novel research.
The english can be improved throughout the manuscript and suggest a detailed proof read.
Examples include:
'In this work, a collaborative experimental study has been conducted in order to assess the effect on the hydrodynamic performance of a compliant composite hydrofoil controlled by an imposed cavity pressure.' should be replaced with 'In this work, a collaborative experimental study has been conducted to assess the effect an imposed cavity pressure has on the controlling the hydrodynamic performance of a compliant composite hydrofoil.'
Usage such as 'figure x shows something' should be replaced with 'something is shown in figure x'.
A major conclusion that is not mentioned is that the morphing of the hydrofoil increases L/D but at the expense of increased minimum pressure and hence the incipient cavitation number. The authors should note this in the discussion and conclusion.
Perhaps the authors should consider rewording the words 'cavity pressure' in the text and title as it is misleading in suggesting the effect of pressure within a vapour cavity on the hydrofoil characteristics rather than and internal pressure. Maybe: Morphing Hydrofoil Model Driven by Compliant
Composite Structure and internal Pressure?
Author Response
Point 1: The english can be improved throughout the manuscript and suggest a detailed proof read.
Response 1: The english has been checked by a native English Colleague.
Point 2: 'In this work, a collaborative experimental study has been conducted in order to assess the effect on the hydrodynamic performance of a compliant composite hydrofoil controlled by an imposed cavity pressure.' should be replaced with 'In this work, a collaborative experimental study has been conducted to assess the effect an imposed cavity pressure has on the controlling the hydrodynamic performance of a compliant composite hydrofoil.'
Response 2: The sentences have been replaced in the text.
Point 3: Usage such as 'figure x shows something' should be replaced with 'something is shown in figure x'.
Response 3: The text has been changed accordingly.
Point 4: A major conclusion that is not mentioned is that the morphing of the hydrofoil increases L/D but at the expense of increased minimum pressure and hence the incipient cavitation number. The authors should note this in the discussion and conclusion.
Response 4: Thank you for this very interesting remark and this comment has been added to the conclusion.
Point 5: Perhaps the authors should consider rewording the words 'cavity pressure' in the text and title as it is misleading in suggesting the effect of pressure within a vapour cavity on the hydrofoil characteristics rather than and internal pressure. Maybe: Morphing Hydrofoil Model Driven by Compliant Composite Structure and internal Pressure?
Response 5: Internal Pressure instead of cavity pressure has been modified.
